# Encoding Hierarchical Information in Neural Networks helps in Subpopulation Shift

## Abstract

Over the past decade, deep neural networks have proven to be adept in image classification tasks, often surpassing humans in terms of accuracy. However, standard neural networks often fail to understand the concept of hierarchical structures and dependencies among different classes for vision related tasks. Humans on the other hand, seem to intuitively learn categories conceptually, progressively growing from understanding high-level concepts down to granular levels of categories. One of the issues arising from the inability of neural networks to encode such dependencies within its learned structure is that of subpopulation shift – where models are queried with novel unseen classes taken from a shifted population of the training set categories. Since the neural network treats each class as independent from all others, it struggles to categorize shifting populations that are dependent at higher levels of the hierarchy. In this work, we study the aforementioned problems through the lens of a novel conditional supervised training framework. We tackle subpopulation shift by a structured learning procedure that incorporates hierarchical information conditionally through labels. Furthermore, we introduce a notion of hierarchical distance to model the catastrophic effect of mispredictions. We show that learning in this structured hierarchical manner results in networks that are more robust against subpopulation shifts, with an improvement up to 3% in terms of accuracy and up to 11% in terms of hierarchical distance over standard models on subpopulation shift benchmarks.

## 1 Introduction

Deep learning has been tremendously successful at image classification tasks, often outperforming humans when the training and testing distributions are the same. In this work, we focus on tackling the issues that arise when the testing distribution is shifted at a subpopulation level from the training distribution, a problem called subpopulation shift introduced recently in BREEDS (Santurkar et al., 2021). Subpopulation shift is a specific kind of shift under the broader domain adaptation umbrella. In domain adaptation, the task of a classifier remains the same over the source and target domains, but there is a slight change in the distribution of the target domain (Goodfellow et al., 2016; Quionero-Candela et al., 2009; Saenko et al., 2010; Ganin & Lempitsky, 2015). In the general setting, the target domain is a slightly changed version of the source domain. For example, an object detector that has been trained to detect objects during day time for a self-driving car application is used to perform the same task, but now on a shifted set of night time images. The task remains the same i.e. to identify and detect objects, but the target domain (night time) is a shifted version of the source domain (day-time), provided all other conditions (such as weather, region, etc.) remain constant. There are other forms of shifts as well such as shifts in the marginal distribution of labels (Tachet des Combes et al., 2020) or shifts under data imbalance (Li et al., 2019). These are broadly denoted as label and target shifts respectively.

However, in the setting of subpopulation shift, both the source and target domains remain constant. The shift here occurs at a more granular level, that of subpopulations. Consider the source distribution described above, that of a self-driving car. Let's say the categories for classification included small vehicles and large vehicles. Under small-vehicles, the source set included samples of golf car and race car, and under

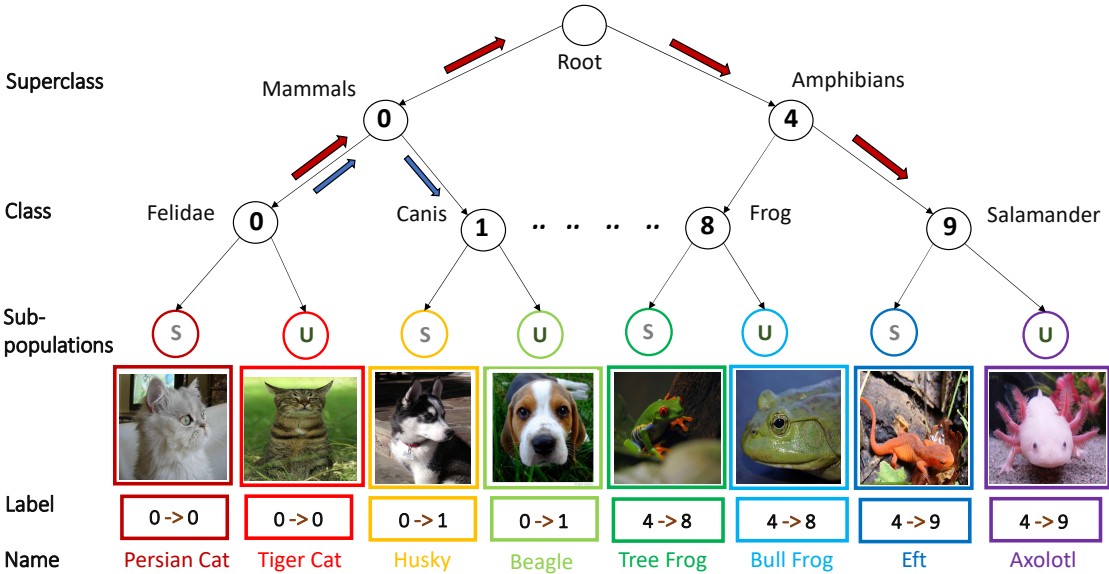

Figure 1: An example hierarchical representation of a custom subset of ImageNet. The classes for the classification task are at the intermediate level, denoted by 'class'. The constituent subpopulations of each class are particular classes from the ImageNet dataset and are marked at the leaf level as 'subpopulations'. The labels for these are not shown to the network. The letter 'S' denotes 'Seen' distribution and 'U' denotes 'Unseen' shifted distributions. One-hot labels are provided at each level of the tree. The colored arrows indicate the hierarchical distance from one leaf node to the other. This shows that mispredicting a Felidae as a Canis (two graph traversals) is less catastrophic than predicting the same as an Salamander (four graph traversals). For illustration we provide the names of one set of subpopulations for each class.

large vehicles, the source samples were from firetrucks and double decker buses. In the target domain for testing, the classes remain unaltered; the classifier is still learning to categorize vehicles into small or large categories. However, the testing samples are now drawn from different subpopulations of each class which were not present during training, such as coupe and sedan for small vehicles and dumpster truck and school bus for large vehicles.

Additionally, the current way of classification, in which each class is considered separate and independent of others, treats the impact of all mispredictions as equal. This is counter-intuitive, since a husky and a beagle are more similar to each other than to a bullfrog. The impact of misclassifications becomes quite important in critical use cases. The cost of mispredicting an animate object for an inanimate object can be disastrous for a self-driving car. To address this, we introduce 'catastrophic coefficient', a quantitative measure of the impact of mispredictions that follows intuitively from a hierarchical graph. It is defined as the normalized length of the shortest path between the true and the predicted classes as per the graphical structure of the underlying hierarchy. We show that incorporating hierarchical information during training reduces the catastrophic coefficient of all considered datasets, under subpopulation shift.

We explicitly incorporate the hierarchical information into learning by re-engineering the dataset to reflect the proposed hierarchical graph, a subset of which is sketched out in Figure 1. We modify the neural network architectures by assigning intermediate heads (one fully connected layer) corresponding to each level of hierarchy, with one-hot labels assigned to the classes at each level individually, as shown in Figure 2. We ensure that only samples correctly classified by a head are passed on for learning to the next heads (corresponding to descendants in the hierarchy graph) by a conditional learning mechanism. We first show

results on a custom dataset we create out of ImageNet, and then scale up to three subpopulation benchmark datasets introduced by BREEDS (Santurkar et al., 2021) that cover both living and non-living entities. We also show results on the BREEDS LIVING-17 dataset by keeping the hierarchical structure, but changing the target subpopulations to cover a more diverse range. We show that given a hierarchy, our learning methodology can result in both better accuracy and lower misprediction impact under subpopulation shift.

- To the best of our knowledge, this is the first attempt to tackle the problem of subpopulation shift by hierarchical learning methods. Our method incorporates hierarchical information in two ways: 1) allowing independent inference at each level of hierarchy and 2) enabling collaboration between these levels by training them conditionally via filtering (Deng et al. (2011). This ensures that each level only trains on samples that are correctly classified on all previous levels. This is similar to anytime inference (Karayev et al. (2014)), but the goal is no longer to enable efficient inference or early exit strategies, but to propagate conditional probabilities.

- Framing the problem in a hierarchical setting allows us to quantify the misprediction impact, measured by the shortest hierarchical distance between the true and predicted labels for inference. While this has been considered in works on cost-sensitive classification (Verma et al. (2012), Bertinetto et al. (2020)), we only use it as an evaluation metric to study the impact of hierarchies on subpopulation shift, instead of directly optimizing it.

- We evaluate the performance of deep models under subpopulation shift and show that our training algorithm outperforms classical training in both accuracy and misprediction impact.

## 2 Related Work

**Subpopulation shift** is a specific variant of domain adaptation where the models need to adapt to unseen data samples during testing, but the samples arrive from the same distribution of the classes, changed only at the subpopulation levels. BREEDS (Santurkar et al., 2021) introduced the problem of subpopulation shift along with tailored benchmarks constructed from the ImageNet (Deng et al., 2009) dataset. WILDS (Koh et al., 2021) provides a subpopulations shift benchmark but for toxicity classification across demographic identities. Cai et al. (2021) tackle the problem through a label expansion algorithm similar to Li et al. (2020) but tackles subpopulation shift by using the FixMatch (Sohn et al., 2020) method. The algorithm uses semi-supervised learning concepts such as pseudo-labelling and consistency loss. Cai et al. (2021) expands upon this and showed how consistency based loss is suitable for tackling the subpopulation shift problem. But these semi-supervised approaches require access to the target set, albeit unlabelled, as the algorithm makes use of these unlabelled target set to further improve upon a teacher classifier. We restrict ourselves to the supervised training framework where we have no access to the target samples. Moreover, we tackle the subpopulation shift problem by incorporating hierarchical information into the models.

**Hierarchical modeling** is a well-known supervised learning strategy to learn semantic concepts in vision datasets. Under this section, we cover methods that are shown on smaller datasets under small-scale methods and works that show results on ImageNet-scale datasets as large-scale methods.

**Large-scale hierarchical methods:Hierarchical modeling** is a well-known supervised learning strategy to learn semantic concepts in vision datasets. Under this section, we cover methods that are shown on smaller datasets under small-scale methods and works that show results on ImageNet-scale datasets as large-scale methods. Yan et al. (2015) introduce HD-CNN, which uses a base classifier to distinguish between coarser categories whereas for distinguishing between confusing classes, the task is pushed further downstream to the fine category classifiers. HD-CNN was novel in its approach to apply hierarchical training for large scale datasets but suffers from a different scalabilty problem. Its training requires copies of network parts for each subtree, and therefore the network size continues to grow with bigger hierarchies. Furthermore, there is sequential pre-training, freezing, training and finetuning required for each level of hierarchy, and hence the authors limit their heirarchies to a depth of 2. Deng et al. (2010) showed that the classification performance can be improved by leveraging semantic information as provided by the WordNet hierarchy. Deng et al. (2014) further introduced Hierarchy and Exclusion Graphs to capture semantic relations between

two labels (parent and children). Although this work relabels leaf nodes to intermediate parent nodes, they train models only on the leaf node labels (single label). Blocks (Alsallakh et al., 2018) visually demonstrates via confusion matrices how learning hierarchies is an implicit method of learning for convolutional neural networks and similar classes are mapped close to one another along the diagonal of the learnt confusion matrix. Song & Chai (2018) shows how multiple heads of a neural network can collaborate among each other in order reach a consensus on image classification tasks. Verma et al. (2020) introduces a dataset with hierarchical labels for Human Pose classification known as Yoga-82 and trains hierarchical variants of DenseNet (Huang et al., 2017) to benhcmark classification accuracy on this set.**In contrast, our work can be extended to multiple levels of hierarchy without the need for changing architecture, while employing a conditional training approach to link multiple labels of a single image as per the provided hierarchy. We show, by utilizing the hierarchy in this manner we are able to mitigate the effect of subpopulation shift, both under accuracy and impact of mispredictions.** Hierarchical inference has often been used to enable efficient inference strategies in the scheme of anytime inference algorithms (Deng et al. (2011); Karayev et al. (2014) ). Additionally, Deng et al. (2011) can also learn a label hierarchy. However with these the aim is to make the inference pipeline more efficient by exiting early for easier examples. We perform inference at all levels of hierarchy to learn coarse-to-fine grained features to help with sub-population shift.

**Small-scale hierarchical methods:** B-CNN (Zhu & Bain, 2017) learns multi-level concepts via a branch training strategy through weighted loss of the individual branches on small-scale datasets. H-CNN (Seo & shik Shin, 2019) leverages hierarchical information to learn coarse to fine features on the Fashion-MNIST (Xiao et al., 2017) dataset. Condition CNN (Kolisnik et al., 2021) learns a conditional probability weight matrix to learn related features to help classification results on Kaggle Fashion Product Images dataset. VT-CNN (Liu et al., 2018) introduces a new training strategy that pays attention to more confusing classes in CIFAR-10 and CIFAR-100 based on a Confusion Visual Tree (CVT) that captures semantic level information of closely related categories. Inoue et al. (2020) show slight improvements on B-CNN by providing hierarchical semantic information to improve fine level accuracy on CIFAR-100 and Fashion-MNIST. CF-CNN (Park et al., 2021) proposes a multilevel label augmentation method along with a fine and several coarse sub-networks to improve upon corresponding base networks. **In our experiments we provide an approach to hierarchically train deep models which scales to ImageNet based subpopulation shift benchmarks.**

**Hierarchical knowledge to make better predictions:** There are works done to learn similarity metrics in the context of hierarchical settings such as in Verma et al. (2012). Another recent work, Bertinetto et al. (2020) introduced a similar notion of impact of mispredictions. Shkodrani et al. (2021) designed a theoretical framework for hierarchical image classification with a hierarchical cross-entropy model to show a slight improvement over Bertinetto et al. (2020). **On the other hand, we use the misprediction distance only as an evaluation metric to quantify the impact of our conditional training framework on the degree of catastrophic predictions, and instead look at the role that hierarchical learning plays in mitigating issues across domain shifts during inference.**

## 3 Methodology: Hierarchies to Mitigate the Effect of Subpopulation Shift

### 3.1 Subpopulation Shift

As described in Section 1, subpopulation shift is a specific branch of the broader domain adaptation problem. In subpopulation shift the training and the testing distributions differ at the level of subpopulations. Let's focus on an n-way classification problem, with each class denoted by $i$; $i = \{1, 2...n\}$. The data consisting of image-label pairs for the source seen and the target unseen domain are denoted by $\{\mathbb{X}^s, \mathbb{Y}^s\}$ and $\{\mathbb{X}^u, \mathbb{Y}^u\}$, respectively. Each class $i$ draws from $s$ different subpopulations. The different subpopulations of class $i$ for training seen domain are denoted by $S_i^s$ and for testing unseen domain by $S_i^u$. We reiterate that between seen and unseen domains, the $n$ classes remain the same, since the classification task is unchanged. However the data drawn for each class at the subpopulation level shifts, with no overlap between the seen and unseen subpopulations. This reflects that the subpopulations used for testing are never observed during training, i.e. $S_i^s \cup S_i^u = \emptyset$.

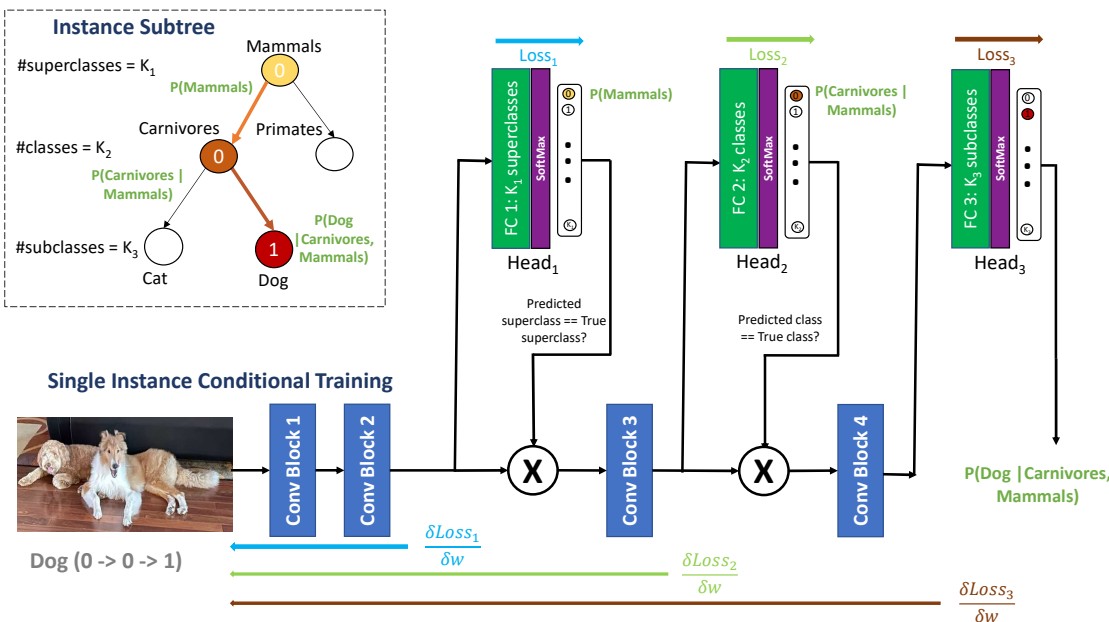

Figure 2: Figure shows our conditional training framework applied to a multi-headed neural network architecture, on the instance subtree shown on the top left. The bottom of the figure shows conditional training for a single instance of a class 'Carnivores', subclass 'Dog'. The shifting subpopulations are located one level below and are not exposed to the training methodology. The conditional training methodology is shown alongside. Conv blocks 1 and 2 make up the backbone that will be used for all 3 heads. We get the superclass prediction from head 1 located after Conv block 2. The multiplier between Conv block 2 and 3 denotes that the output of Conv block 2 only passes to Conv block 3 if the prediction of head 1 (i.e. the superclass) is correct. If head1 predicts the incorrect superclass, the rest of the network does not train on the instance. Similarly, head 2 predicts the class at the next hierarchy level, and dictates whether the fourth Conv block will be trained on this instance or not. The blocking or passing of the instance to different parts of the architecture is implemented in a batch setting via the validity mask, described in Figure 3.

## 3.2   Hierarchical View to tackle Subpopulation Shift

We tackle the subpopulation shift problem by explicitly incorporating hierarchical knowledge into learning via labels. Intuitively, if a neural network can grasp the concept of structural hierarchies, it will not overfit to the observed subpopulations. Instead, the network will have a notion of multiple coarse-to-fine level distributions that the subpopulation belongs to. The coarser distributions would likely cover a much larger set of distributions, hopefully helping in generalization under shift. For instance, a network trained with the knowledge that both a fire-truck and a race-car fall under vehicles, and a human and a dog fall under living things, will not overfit to the particular subpopulation but have a notion of vehicles and living things. This will allow it to generalize to a newer large-vehicle such as school-bus and predict it as a vehicle rather than a living thing, since the network has learned a much broader distribution of vehicles one level of hierarchy above. Even if there is a misprediction, it is more likely to be at the lower levels of hierarchy, confusing things that are less catastrophic to mispredict.

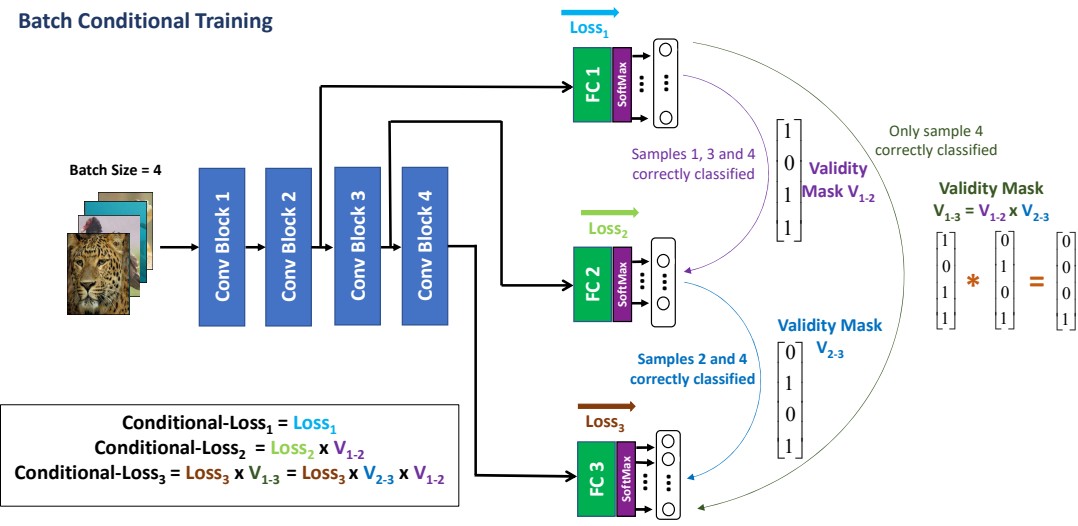

Figure 3: Practical implementation of conditional training for a batch of images. The validity mask serves to ensure that the blocks corresponding to a particular level are trained only on the instances that are correctly classified at the previous level. Instead of blocking representations by multiplying with zeros as shown in Figure 2, we implement conditional training via multiplication of losses with the corresponding validity masks, resulting in the same outcome. Validity masks $V_{l_1-l_2}$ represent the propagation of correctly classified instances from level $l_1$ to $l_2$, and contain a 1 where the instance was correctly classified by all levels between $l_1$ and $l_2$ and 0 otherwise. They can be built from the composition of several validity masks. For instance, as shown in the figure, the validity mask for propagation from level 1 to level 3 is calculated by multiplying the validity mask from level 1 to level 2 with the validity mask from level 2 to level 3.

### 3.3 Vision Datasets as Hierarchical Trees

ImageNet (Deng et al., 2009) is a large-scale image database collected on the basis of an underlying hierarchy called WordNet (Miller, 1992). It consists of twelve different subtrees created by querying synsets from the WordNet hierarchy. To motivate the problem of subpopulation shift, we create two custom datasets from ImageNet, which are shifted versions of each other at the subpopulation level. The datasets have a balanced hierarchical structure of depth 3 as shown in Figure 1, starting from coarse concepts such as mammals and amphibians at a higher level, to fine grained specific subpopulations at the leaf nodes.

The hierarchical structure has 5 nodes at the highest level of superclasses, 10 nodes at the class level ($n = 10$), and each class draws from 3 subpopulations each ($s = 3$), leading to a total of 30 subpopulations per dataset. Each subpopulation is a class from ImageNet. Figure 1 shows a partial hierarchy from the dataset, showing one out of the three subclasses for seen and unseen datasets at the leaf nodes per class. This is a ten-way classification task. Each class consists of shifting subpopulations, shown one level below. During testing under shift, the 10 classes at the class level remain the same, but the 30 subpopulations that samples are drawn from are changed.

Given a tree, we start at the root node and traverse downwards to the first level of hierarchy, which consists of superclasses such as mammals, fish, reptiles, etc. The custom dataset, for instance, has five superclasses, labelled $0 - 4$. Next we traverse to the level of classes. These are the actual tasks that the network has to

classify. At this level, finer concepts are captured, conditioned on the previous level. For instance, the task now becomes: given an amphibian, is it a frog or a salamander; or given a bird, is it aquatic or aviatory. Each superclass in our custom dataset has only 2 classes, making up the $n = 10$ classes for classification. This level has one-hot encoding of all ten classes. Thus, the categorical labels are presented in a **level-wise concatenated format** as shown in Figure 1. The label for frog is '4 → 8', with the label 4 encoding that it belongs to the superclass of amphibians and 8 encoding that conditioned on being an amphibian, it is a frog. The models only see labels till the class level; the subpopulations labels are hidden from the networks. Finally, we reach the leaf nodes of the tree, where there are three subpopulations per class (figure only shows 1 from seen and unseen distributions). This overall encoding represents each label as a path arising from the root to the classes. The class labels always occur at $level = depth - 1$, one level above the subpopulations. For datasets such as LIVING-17 with a $depth = 4$, classes occur at $level = 3$ and we show an instance of this hierarchy in Figure 2.

Accuracy and catastrophic coefficients are reported for the 10 classes, similar to BREEDS. The custom trees are simple and balanced, capturing the hierarchical structure found in the dataset. We use them to lay the foundations on which we implement our conditional training framework. The two custom datasets are flipped versions of each other, created by keeping the hierarchical structure fixed. In one dataset, one subpopulation set becomes the seen distribution whereas the other one becomes the unseen one, and this is reversed for the second dataset. We then show how our method translates well to complicated hierarchies such as the LIVING-17, Non-LIVING-26, and ENTITY-30 (Santurkar et al., 2021) subpopulation shift benchmarks. This illustrates that our algorithm is compatible with any hierarchy chosen according to the task of interest.

### 3.4 Catastrophic Distance

In this section, we introduce the concept of catastrophic coefficient as a measure of the impact of misprediction. It is the shortest hierarchical distance between the true label and the predicted label in our hierarchy, normalized by the number of samples. It implicitly quantifies whether there is a notion of semantic structure in the model's predictions. Subpopulation shift occurs at a lower level of a hierarchical tree where unseen subclasses are introduced during evaluation. So, if the hierarchically trained networks can grasp the concepts of superclasses and classes, the mispredictions during the shift will not be catastrophic. This is because they will tend to be correct at the higher levels, and hence 'closer' to the ground truth node in terms of graph traversal.

Neural networks trained via standard supervised learning have no explicit knowledge of inter-class dependencies. Thus, for flat models, mispredicting a specific sub-breed of a dog as a sub-breed of a cat is as catastrophic as mispredicting the same as a specific species of a snake. hierarchical distance between the true and predicted classes intuitively captures the catastrophic impact of a misprediction and accounts for the semantic correctness of the prediction. This serves as an additional metric to accuracy for evaluating the performance of models under subpopulation shifts. Additionally, it illustrates that the improvement in accuracy is truly due to incorporating better hierarchical information, rather than model architecture changes or the conditional training framework. It is pictorially illustrated by the colored arrows in Figure 1.

A higher hierarchical distance between a misprediction and its ground truth signifies a more catastrophic impact. We average the graph traversal distances of all predictions ($= 0$ if sample classified correctly) over the entire dataset and call it the catastrophic coefficient, thus quantifying the impact of mispredictions for a network-dataset pair. Formally, let $g_k$ be the graph traversals needed for sample k in the shortest path between its true and predicted label. Let there be $N$ samples for evaluation. Then, the catastrophic coefficient is defined as $Cat = \frac{\sum_{k=1}^{N} g_k}{N}$. We note that we use this distance just to evaluate, and not during training. For evaluation, we take the final classifier level predictions and run it via our graph to check for distances, irrespective of whether they have been shown the hierarchy or not.

### 3.5 Architecture

We modify the standard ResNet (He et al., 2016) architectures to make them suitable for our conditional training framework. Since our network makes classification decisions at each level of the hierarchy, we

Table 1: Details of the Subpopulation Shift Datasets

| Datasets | Depth | Subpopulations (s) | Classes (n) |
|----------|-------|--------------------|-------------|
| Custom | 3 | 3 | 10 |
| LIVING-17 | 4 | 2 | 17 |
| Non-LIVING-26 | 5 | 2 | 26 |
| ENTITY-30 | 5 | 4 | 30 |

introduce a separate head to predict the one-hot encoded vectors at each level. In a hierarchical subtree, the concept of a dog class is is represented as a mammal at the superclass level, a carnivore at the class level and a dog at the subclass level. We want to train a multi-headed network where each head is trained on level wise concepts starting from the superclass level, all the way down to the subclass level maintaining the path in the subtree. Thus if we want to represent the hierarchical concept of a dog as shown in Figure 2, we want the $Head_1$ of our network to predict if it is a mammal (superclass), $Head_2$ to predict if it is a carnivore (class) and finally $Head_3$ to predict that it is a dog (subclass). Convolutional Neural Networks learn coarse to fine features as they go deeper, capturing local concepts of images in the early layers and global concepts in the later layers (Zeiler & Fergus, 2014). This lines up well with our hierarchical structure, and hence we connect the different heads at different depths of the model. The concept is pictorially depicted in Figure 2. Since we use Residual Networks in our work, the individual convolutional blocks here are residual blocks. The locations of these heads are determined experimentally. We got best results with $Head_1$ attached after the third residual block, $Head_2$ and $Head_3$ after the fourth residual blocks for the subtree shown in Figure 2. We further want to ensure collaboration between these heads, done via a conditional training approach which we describe next.

### 3.6 Conditional Training Details

Here we describe the conditional training framework, illustrated in Figure 3, which is independent of subpopulation shift. Let us assume we have 3 levels in our hierarchy, with the levels enumerated by $l = 1, 2, 3$. Let the labels at each of these levels (treated as one-hot) be denoted by $y_l$. Let $F$ be the neural network that we pass a batch of images $X$ to. Here $X \in \mathbb{R}^{B \times d}$ where B is the batch-size and d is the dimension of the input data. Further, let $F_l$ be the neural network up to head $l$, corresponding to predicting at level $l$ of our hierarchical graph. Note that all $F_l$ have overlap since the neural network is shared, rather than an ensemble, as illustrated in Figure 2. The conditional loss at head $l$, $L_l$ is calculated as:

$$L_l = CrossEntropy(F_l(x), y_l) * (V_{1-l})$$

where $V_{1-l}$ is the validity mask. $V_l \in R^B$ contains all zeros, except ones at locations where the samples have been correctly classified by all heads until the head at level $l$. Each $V_l$ is generated by element-wise multiplication of the individual constituent masks, $[V_{1-2} * V_{2-3} * ... * V_{l-1-l}]$, allowing for an incorrect classification at any head to block further propagation of the incorrect instance to all deeper heads, shown pictorially in Figure 3. This validity mask is what enforces our conditional training framework. Multiplying this validity mask with the current head's loss before backpropagation ensures that learning for $F_l$ only occurs on samples that are meaningful at that head. In other words, $V$ propagates only correctly classified samples, as per its name. This ensures that the prediction at the $l - th$ head is not just $p(y_l)$, but $p(y_l \mid F_k(x) = y_k) \; \forall k = 1, 2..., l-1$. In other words, it allows each head's outcome to represent the probability of the current head's prediction given that prediction of all levels until the current one were correct, allowing the network to learn progressively refining predictions. The network until a particular head is trained progressively on the conditional loss corresponding to that head. That means that during backpropagation, each layer get gradients from all conditional losses of heads located after that layer. This allows us to learn a shared backbone, but progressively refine features from coarser to finer as pertaining to the hierarchy.

## 4 Experiments & Results

In Section 3, we described in detail our conditional training framework, where we train multi-headed networks to incorporate the notion of hierarchies in vision datasets. In this section, we empirically demonstrate how models trained with our approach perform better under subpopulation shift than models trained in a traditional flat learning setup. As a proof of concept, we first show results on two custom datasets created by querying the ImageNet on living entities. We then show the efficacy of our approach by expanding to three subpopulation shift benchmarks introduced in BREEDS (LIVING-17, Non-LIVING-26 and ENTITY-30). Each of these subpopulation shift benchmarks have varying structures in terms of depth and width and each captures a diverse set of relationships among their entities. We explain each benchmark in details in the following, corresponding subsections. We compare our approach with a baseline model trained in the classical manner on all classes without any hierarchical information. We additionally compare with Branch-CNN (Zhu & Bain, 2017), trained as per the branch training strategy outlined by the authors. In Section 2, we mentioned HD-CNN (Yan et al., 2015) in terms of its novelty in training hierarchical deep models but as mentioned, it suffers from the issue of scalability and memory footprint with expanding hierarchies. The method is limited to hierarchies of depth 2, whereas each subpopulation benchmark exhibits trees of depth 3 or more. The method requires training one coarse classifier and multiple fine classifiers depending on how many coarse categories there are. With the current architectures of deep models, having multiple pretrained coarse and fine classifiers will vastly increase memory footprint and training time. Hence, we do not compare with H-CNN. In terms of both accuracy and catastrophic coefficient, **we show that our hierarchical models are superior to baseline class models and Branch-CNN in tackling the subpopulation shift problem in all the five cases considered.** We note that in 3 out of 5 sets of results, the trend of improvement in the subpoplation shifted (unseen) dataset was tracked by the unshifted (seen)) dataset as well. However, the trend is not unanimous. For instance, improvements in accuracies track each other roughly in both custom datasets and LIVING-17 datasets, but not for non-LIVING-26 and ENTITY-30 datasets. We believe that the kind of shifted subpopulation itself has an impact on this, as evidenced by the different results we get by shifting one source to three targets in LIVING17-A, B and C. We also believe that the classes where the shift occurs determine how easy the categorization under shift is, and hence see different trends for say, LIVING-17 and Non-LIVING-26 datasets.

### 4.1 Overall Setup

As mentioned, we consider subpopulation shift one level below the class level of a hierarchy. In this section we briefly describe the setup with the custom datasets as example. We provide the exact details of each benchmark in the subsequent sections, summarized in Table 1. Consider an n-way classification problem, with each class denoted by $i$; $i = \{1, 2...n\}$. For our custom datasets, $n = 10$. The total number of levels of hierarchy including the subpopulation levels, $l$, is 3. The $n$ classes are located at $l = 2$ in our custom tree. Now, we create the shift by sampling subpopulations of each class $i$ from $s$ different subpopulations. For the custom datasets, $s = 3$ and thus, for the custom datasets we have a 10-way classification problem with a total of $n \times s = 30$ subpopulations. More concretely, the subpopulations for class $i$ are distributed over $S_i^s$ (seen) and $S_i^u$ (unseen) domain. Let's consider $i = \{dogs\}$, $S_{dogs}^s = $ [Bloodhound, Pekinese] and $S_{dogs}^u = $ [Great-Pyrenees, Papillon]. Thus the learning problem is that by training on just the seen subpopulations $S_{dogs}^s$, the model should be able to identify that the unseen subpopulations of $S_{dogs}^u$ belong to $i = \{dogs\}$.

We use accuracy and catastrophic coefficient described in Section 3.4 to measure performance, both in the presence and absence of subpopulations shift. The higher the accuracy of a model, the better it is. On the contrary, the lower the catastrophic co-efficient the better it is for a model. The number of graph traversals from the predicted node to the ground-truth node represents the value of a single misprediction impact, which varies from a minimum value of 0 (correct prediction) up to a maximum value of $2 \times (depth - 1)$ (worst case prediction, where predictions are made one level above subpopulations, and hence at a level of $(depth - 1)$). For example, for a dataset with $depth = 4$ such as LIVING-17 the worst case misprediction value for a single instance is 6, whereas for Non-LIVING-26, the same is 8. Both accuracy and catastrophic coefficient are reported mainly under two different settings, differentiated by the subscript. The prefix of the subscript determines the domain the model was trained on and the suffix denotes the domain it is evaluated on. There are two combinations, '$s - s$' and '$s - u$', with 's' representing seen data and 'u' representing

Table 2: Results on Custom Dataset 1 (left) and Custom Dataset 2 (right). Corresponding catastrophic coefficients are shown in the bar plot on the right.

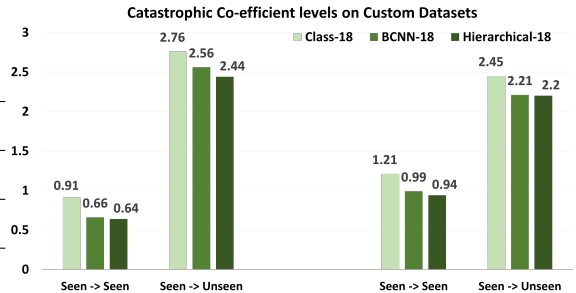

| Model | $Acc_{s-s}$ | $Acc_{s-u}$ | $Acc_{s-s}$ | $Acc_{s-u}$ |
|---|---|---|---|---|
| Baseline-18 | 83.47 | 48.69 | 77.73 | 55.0 |
| BCNN-18 | 87.87 | 51.33 | 81.73 | 58.96 |
| Hierarchical-18 | **88.27** | **53.76** | **82.48** | **59.35** |

Figure 4: Results on Custom Datasets 1 and 2. Accuracy is shown on the left, and corresponding catastrophic coefficients on the right. Our model outperforms the other in both accuracy and catastrophic coefficient on both 's-s' and 's-u' populations.

unseen data. '$s - s$' does not evaluate subpopulation shift, but shows the results of using our method as a general training methodology. It is trained on the standard training data of the seen domain, and evaluated on the validation set in the same seen domain. '$s - u$' evaluates results under subpopulation shift: training is performed on seen domain, and testing on unseen domain. Details of hierarchies and code will be made available soon.

## 4.2 Model Setup

Throughout our experiments we focus mainly on three sets of training, which result in the Baseline, the Branch-CNN and the Hierarchical Models. The Baseline models are trained in a flat manner, and evaluated as per the hyper-parameters and training details as mentioned in BREEDS (Santurkar et al., 2021), except bootstrapping. The classification task is on the $i$ classes, enumerated at the level of **'classes'** mentioned in the hierarchy. The subpopulation shift occurs one level below. The subpopulations labels are never shown to the network. The Hierarchical and Branch-CNN models, have been trained on the complete hierarchical information present in the tree, using our conditional training framework and the Branch Training Strategy (Zhu & Bain, 2017) respectively. The method oversees training to teach coarse to fine concepts as per the hierarchical structure of the target classes. The method invokes a weighted summation of a joint loss where each loss contribution comes from each branch (head) of a multi-headed network. There is a loss weight factor associated with each branch which dictates how much of each branch contributes towards the final loss as training continues. A head in our case is analogous to a branch in theirs. A branch represents a conceptual level in the hierarchy, and the training scheme dictates how much weight each branch contributes to the weighted loss as training goes on. In our conditional training framework, we sequentially train each head of our multi-headed network with the subsequent conditional loss as discussed in Figure 2. In this manner we teach the top-down hierarchical structure to networks and the conditional taxonomic relationships among its entities.

We use ResNet-18 as our network architecture backbones for modifications as mentioned in subsection 3.5. For enumerating results, we use 'Hierarchical-18' to denote a modified ResNet-18 model trained conditionally. Similarly 'Baseline-18' signifies a ResNet-18 architecture trained for flat classification on the $n$ categories found at the level of 'classes' in the hierarchy. BCNN-18 refers to the modified ResNet-18 architecture trained via the Branch Training Strategy (Zhu & Bain, 2017).

## 4.3 Results on Custom Datasets

In this section, we discuss the results on the two custom datasets, shown in Figure 4. We train the models on each dataset for 120 epochs with a batch size of 32, starting with a learning rate of 0.1, and a 10 fold drop every 40 epochs thereafter. We do not use data augmentation on our custom datasets. All models have

Table 3: Results on LIVING-17, with and without shift is shown on the left. Results for shift on Living-17-B and Living-17-C are shown as well. Corresponding catastrophic coefficients are shown in the bar plot on the right.

| Model | $Acc_{s-s}$ | $Acc_{s-u}$ | $Acc_{s-u}(B)$ | $Acc_{s-u}(C)$ |
|---|---|---|---|---|
| Baseline-18 | 92.3 | 57.02 | 53.54 | 53.04 |
| BCNN-18 | 92.88 | 58.8 | 55.66 | 55.1 |
| Hierarchical-18 | **93.17** | **60.53** | **56.6** | **55.62** |

**Catastrophic Co-efficient Levels on LIVING-17, -B and -C**

Figure 5: Results on Living-17. Accuracy is shown on the left, and corresponding catastrophic coefficients on the right. Additional experiments for shift on 2 variants, Living-17-B and -C are also shown. Our model outperforms the others in both accuracy and catastrophic coefficient on both 's-s' and 's-u' populations, including shifted performance on the -B and -C variants.

been trained on three random seeds each and the mean numbers are reported. $Acc_{s-u}$ and $Cat_{s-u}$ denote the accuracy and catastrophic coefficient of the model during the $s - u$ shift at the class level. As shown in Figure 4, **our Hierarchical-18 model performs better than the others, both in terms of accuracy and catastrophic coefficient, as well as both in the presence and absence of subpopulation shift.** Moreover, the performance gap is significant under shift indicating that the imparted hierarchical information is helpful in correctly predicting unseen subpopulation classes. The models trained conditionally have $\sim 4 - 5\%$ improvement in terms of accuracy and an improvement of $\sim (0.25 - 0.31)$ in hierarchical distance, translating into 10.0% and 7.9% improvement in terms of catastrophic coefficient over the flat baseline class level models under shift for the two custom sets. Figure 4 highlights another interesting fact. Both the custom datasets have the same hierarchical structure and model the same semantic relationships among the entities; the difference is created by populating each set with different subpopulations. All models suffer performance drops from custom set 1 to 2, showing the adverse effects of the distribution spanning a particular set, implying that some subpopulation shifts are just inherently harder to tackle.

### 4.4 Results on LIVING-17

In this section we discuss the results on the BREEDS LIVING-17 dataset, enumerated in Figure 5. For the LIVING-17 dataset, $n = 17$, depth is 4 and $s = 2$. The $n$ classes are located at depth $l = 3$ and the subpopulations at $l = 4$ respectively. This subpopulation shift benchmark, introduced in BREEDS (Santurkar et al., 2021), captures finer details of hierarchy that encode richer relationships between the entities. We show that our methodology is applicable to complex hierarchies and outperforms class level baseline and BCNN-18 models both on $Acc_{s-u}$ and $Cat_{s-u}$. We train each architecture and model on five random seeds and report the mean numbers. We report numbers without bootstrapping, but follow all their other hyper-parameters reported by BREEDS. Under the shift, Hierarchical models achieve $\sim 1.7 - 3.5\%$ and $0.17 - 0.20$ improvement in terms of accuracy and hierarchical distance respectively over other techniques. This results around 4% and 11% in terms of catastrophic coefficient over BCNN-18 and Baseline-18 respectively, as seen in Figure 5.

**Results on Shifted LIVING-17** To cover a more diverse shift, we retain the hierarchy introduced in LIVING-17 but consider 2 more sets of different subpopulations. We call these LIVING-17-B and LIVING-17-C. These two shifted versions of the unseen set of LIVING-17 are formed by varying the $S_i^u$ subclasses. We do this either by adding disjoint subclasses of the ImageNet (Deng et al., 2009) or by creating different combinations of the existing $S_i^u$ with new disjoint subclasses. We reuse some of the $S_i^u$ subclasses due to the unavailability of the same in the ImageNet database. All the subpopulations of $i = \{wolf\}$ from the

Table 4: Results on Non-LIVING-26, with and without shift. Corresponding catastrophic coefficients are shown in the bar plot on the right. The L3 Hierarchy is the same hierarchy with the first two levels collapsed into a single level.

| Model | $Acc_{s-s}$ | $Acc_{s-u}$ |
|---|---|---|
| Baseline-18 | **88.39** | 42.13 |
| BCNN-18 | 88.1 | 42.4 |
| L3 Hierarchical-18 | 87.71 | 42.44 |
| Hierarchical-18 | 87.41 | **42.94** |

**Catastrophic Co-efficient Levels on NON-LIVING-26**

Figure 6: Results on Non-LIVING-26 dataset. Accuracy is shown on the left, and corresponding catastrophic coefficients on the right. Our model outperforms the others in both accuracy and catastrophic coefficient on both 's-u' evaluation, but shows slightly worse performance on 's-s' evaluation.

ImageNet database have already been covered in the $S^s_{wolf}$ and $S^u_{wolf}$ set, so we just reuse the $S^u_{wolf}$ in the sets B and C.

$Acc_{s-u}(B)$ denotes the model accuracy for the shift 's − u' from set A to B. As can be seen from Figure 5, **Hierarchical-18 models have better accuracy and catastrophic coefficients than the other models for all three shifted sets.** This shows that imparting hierarchical knowledge helps deep models to adapt to various degrees of the subpopulation shift.

### 4.5 Results on Non-LIVING-26

In this section, we describe results on the Non-LIVING-26 dataset, tabulated in Figure 6. The dataset has $n = 26$ classes, a depth of 5 and number of subpopulation, $s = 2$. The $n$ classes are located at depth $l = 4$ and the subpopulations at $l = 5$ respectively. For comparison, we train Baseline-18 and BCNN-18. We know that depth might hinder our conditional training process, since we limit samples that pass down from a level to the next contingent on their correct prediction at that head. To test this, we create a collapsed version of this hierarchy. We collapse levels 1 and 2 into a single level and create a new hierarchy with the same amount of information and term this as L3 Hierarchical-18. All models are trained on three random seeds each and the mean numbers are reported.

As seen from Figure 6, **BCNN-**18 **outperforms our hierarchical model on '$s-s$' performance, while our framework performs better under both kinds of shift.** In our conditional training framework, we only train subsequent heads if the previous heads have correctly classified the sample. As the depth of the hierarchical tree increases, fewer samples reach the final head for training, affecting the final classification performance on '$s - s$' models. Despite that, we outperform aseline-18 and BCNN-18 both in terms of accuracy and catastrophic co-efficient on the 'subpopulation shift '$s - u$' set. Since, the L3 Hierarchical-18 model is trained on one less level of hierarchical information, the final head gets to classify some more samples than Hierarchical-18 and has slightly better '$s - s$' performance. We evaluate the catastrophic co-efficient of each model under two different settings. As the name suggests $Cat(3)_{s-s}$ quantifies the effect of catastrophic mispredictions calculated on the collapsed L3-Hierarchy. The BCNN-18 model was trained with all four levels of hierarchical information. Yet, under '$s - u$', the L3 Hierarchical-18 model performs slightly better than the former, which shows the benefits of our conditional training framework.

Table 5: Accuracy results on ENTITY-30, with and without shift. The networks are trained on a collapsed hierarchy of 2 levels, but catastrophic coefficients are evaluated on both the collapsed and original, uncollapsed hierarchy of levels 2 and 4, respectively, shown in brackets on the right

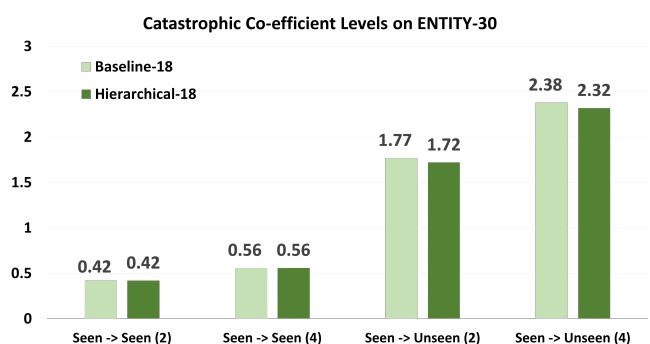

| Model | $Acc_{s-s}$ | $Acc_{s-u}$ |
|---|---|---|
| Baseline-18 | **87.98** | 49.52 |
| Hierarchical-18 | 87.93 | **50.24** |

Figure 7: Results on training networks on the collapsed version of ENTITY-30. Accuracy is shown on the left, and corresponding catastrophic coefficients on the right. Our model outperforms the flat baseline in both accuracy and catastrophic coefficient on both the collapsed and un-collapsed versions of ENTITY-30 under shift.

### 4.6 Results on ENTITY-30

We saw the effect of collapsing hierarchy with the previous set of experiments on Non-LIVING-26. Now, we attempt to understand the results of doing the reverse. In this case, we endeavor to answer that if we train a model on the collapsed version of a hierarchy, would the model still perform better on the original un-collapsed hierarchy that it did not get to see. To perform this experiment, we train on a collapsed version of the ENTITY-30 dataset and test on both the collapsed version and the un-collapsed (original) version. The dataset has $n = 30$, a depth of 5 and $s = 4$. The $n$ classes are located at depth $l = 4$ and the subpopulations at $l = 5$ respectively. The hierarchical tree encapsulates both living and non-living entities and the more meaningful information is embedded between levels 3 and 4. Hence, we collapse the hierarchical information from levels $1 - 3$ to a single level. We train the Hierarchical-18 models on these two levels only and the Baseline-18 models are trained flat on all the classes. All models have been trained on three random seeds each and the mean numbers are reported in Figure 7. The catastrophic coefficients are reported for '$s - s$' and '$s - u$' cases with the number of levels for evaluation in the hierarchy in brackets. To summarize, the networks are trained on 2 levels, but evaluated additionally on an expanded 4 level hierarchy. We note that the Hierarchical-18 has a comparable performance with Baseline-18 on '$s - s$' set but on the shifted unseen distribution, there is a boost in both accuracy and catastrophic co-efficient. **Under both the collapsed and expanded hierarchies, our models have has less catastrophic mispredictions under both 's-s' and 's-u' settings.**

## 5 Conclusion

In this paper, we target the problem of subpopulation shift, which is a specific kind of shift under the broader umbrella of domain adaptation. The subpopulations that make up the categories for the classification task change between training and testing. For instance, the testing distribution may contain new breeds of dogs not seen during training, but all samples will be labeled 'dog'. We note an implicit notion of hierarchy in the framing of the problem itself; in the knowledge of all constituent subpopulations sharing the common immediate ancestry. In line with this, we extend the notion of hierarchy and make it explicit to better tackle the issue of subpopulation shift. We consider the underlying hierarchical structure of vision datasets, in the form of both our own custom subsets and benchmark datasets for subpopulation shift. We incorporate this information explicitly into training via labeling each level with an individual one-hot label, and then encourage collaboration between multiple heads of a model via a conditional training framework. In this framework, each head is only trained on samples that were correctly classified at all levels before the present

one. We further introduce a metric to capture the notion of semantic correctness of predictions. It uses the shortest hierarchical distance between the misprediction and the true label as per the hierarchy to quantify the catastrophic impact of mispredictions. We show that our hierarchy-aware conditional training setup outperforms flat baselines by around $\sim (1-5)\%$ in terms of accuracy and $\sim (3-11)\%$ in terms of catastrophic coefficient over standard models across two custom datasets and three subpopulation shift benchmarks.

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

# A    Appendix

We mention some additional related literature in this section.

## A.1    Additional Related Work

**Hierarchy based Semantic Embedding.** DeVise (Frome et al., 2013) presents a deep visual-semantic embedded model which learns similarity between classes in the semantic space both from images as well as unannotated text. Barz & Denzler (2019; 2020) demonstrates how prior knowledge can be leveraged based on hierarchy of classes such as WordNet to learn semantically discriminating features. Such learnt class embeddings projected on a unit hypersphere proved to be beneficial for both novel class predictions as well as image retrieval tasks.

**Domain Adaptation** and its variants are a well studied set of problems in deep learning. One direction of works (Ben-David et al. (2006), Saenko et al. (2010), Ganin & Lempitsky (2015), Courty et al. (2017), Gong

et al. (2016), Donahue et al. (2014), Razavian et al. (2014)) is aimed at tackling the problem of adapting to target domains by learning on a selective set of samples from the target domain itself. Another line of work aims to match the source and target distributions in the feature space (Glorot et al. (2011), Ajakan et al. (2014), Long et al. (2015), Ganin et al. (2016)). The main motive behind these set of works is to tackle out-of-support domain adaptation tasks by sharing a common representation between the two. To adapt to newer environments, deep models are trained gradually to make them more suitable for transition to these newer environments (Gopalan et al. (2011), Gong et al. (2012), Glorot et al. (2011), Kumar et al. (2020), Chen et al. (2020)). Domain generalization enables the use of multiple different environments during training, but requires having a prior knowledge on the target distribution (Ghifary et al. (2015), Li et al. (2018), Arjovsky et al. (2019), Ye et al. (2021)). We on the other hand, focus on a more specific problem of distribution shift, wherein the shift occurs at a subpopulation level in the target domain.

**Hierarchical Learning for in Non-Supervised Approaches.** Tree-CNN (Roy et al., 2020) tackles the incremental learning problem where the model expands as a tree to accommodate new classes. Zheng et al. (2017) and Qu et al. (2017) tackle the problem of metric learning via hierarchical concepts on large scale image datasets. Chen et al. (2019) applies a semi-supervised approach to learn cluster level concepts at higher level of a hierarchy and categorical features at leaf node levels. Jiang et al. (2019) proposes a Conditional class-aware Meta Learning framework that conditionally learns better representations through modeling inter-class dependencies. Seo & Kim (2019) incorporates a hierarchical semantic loss function together with a confidence estimator to improve performance of zero-shot learning in terms of hit@k accuracy. Works such as McClelland et al. (2016) and Saxe et al. (2013) tried to understand the importance of hierarchical learning from a theoretical perspective and demonstrated an implementation on a neural network based model.

**Hierarchical Learning for Interpretability.** Interpreting predictions from CNNs has been key in understanding what features models look at in order to make predictions. Zhang et al. (2019) provide a semantic as well as quantitative explanations for CNN predictions based on a decision tee in a coarse-to-fine manner at different fine-grained levels. Building on this concept, Hase et al. (2019) introduces a model that leverages a predefined taxonomy to explain the predictions at each level of the taxonomy essentially showing how a Capuchin is gradually classified first as an animal, followed by a primate and finally as a Capuchin as per the hierarchy.

**Applications of Hierarchical Learning.** Dhall et al. (2020), Dhall (2020) show how an image classifier augmented with hierarchical information based on entailment cone embeddings outperforms flat classifiers on an Entomological Dataset. Pham et al. (2021) takes advantage of the relationship between diseases in chest X-rays to learn conditional probabilities through image classifiers. Taoufiq et al. (2020) adapts a similar approach to learn urban structural relationships.

## A.2 Experiments

### A.2.1 Custom Datasets

Results on Custom Dataset 1 (left) and Custom Dataset 2 (right). Mean and standard deviations are reported for five random trials.

| Model | $Acc_{s-s}$ | $Acc_{s-u}$ | $Acc_{s-s}$ | $Acc_{s-u}$ |
|---|---|---|---|---|
| Baseline-18 | $83.47 \pm 0.95$ | $48.69 \pm 1.32$ | $77.73 \pm 2.01$ | $55.0 \pm 0.87$ |
| BCNN-18 | $87.87 \pm 0.71$ | $51.33 \pm 1.22$ | $81.73 \pm 0.51$ | $58.96 \pm 1.47$ |
| Hierarchical-18 | $\mathbf{88.27} \pm 0.88$ | $\mathbf{53.76} \pm 0.47$ | $\mathbf{82.48} \pm 0.54$ | $\mathbf{59.35} \pm 1.65$ |

### A.2.2 LIVING-17

Results on LIVING-17, with and without shift is shown on the left. Results for shift on Living-17-B and Living-17-C are shown as well. Mean and standard deviations are reported for five random trials.

| Model | $Acc_{s-s}$ | $Acc_{s-u}$ | $Acc_{s-u}(B)$ | $Acc_{s-u}(C)$ |
|---|---|---|---|---|
| Baseline-18 | $92.3 \pm 0.84$ | $57.02 \pm 1.48$ | $53.54 \pm 2.28$ | $53.04 \pm \pm 1.9$ |
| BCNN-18 | $92.88 \pm 0.29$ | $58.8 \pm 0.51$ | $55.66 \pm 0.52$ | $55.1 \pm 0.96$ |
| Hierarchical-18 | $\mathbf{93.17} \pm 0.34$ | $\mathbf{60.53} \pm 0.89$ | $\mathbf{56.6} \pm 0.96$ | $\mathbf{55.62} \pm 0.82$ |

### A.2.3   Non-LIVING-26

Results on Non-LIVING-26, with and without shift. The L3 Hierarchy is the same hierarchy with the first two levels collapsed into a single level. Mean and standard deviations are reported for five random trials.

| Model | $Acc_{s-s}$ | $Acc_{s-u}$ |
|---|---|---|
| Baseline-18 | $\mathbf{88.39} \pm 0.32$ | $42.13 \pm 0.93$ |
| BCNN-18 | $88.1 \pm 0.43$ | $42.4 \pm 0.28$ |
| L3 Hierarchical-18 | $87.71 \pm 0.16$ | $42.44 \pm 0.31$ |
| Hierarchical-18 | $87.41 \pm 0.34$ | $\mathbf{42.94} \pm 0.46$ |

### A.2.4   ENTITY-30

Accuracy results on ENTITY-30, with and without shift. Mean and standard deviations are reported for five random trials.

| Model | $Acc_{s-s}$ | $Acc_{s-u}$ |
|---|---|---|
| Baseline-18 | $\mathbf{87.98} \pm 0.1$ | $49.52 \pm 0.16$ |
| Hierarchical-18 | $87.93 \pm 0.09$ | $\mathbf{50.24} \pm 0.26$ |

