# OpenReview forum: "Encoding Hierarchical Information in Neural Networks \\helps in Subpopulation Shift"
_TMLR — Rejected by TMLR_

### Review · Reviewer_Z8z2 · 2022-07-20

**Summary Of Contributions:**

This paper proposed a new training paradigm to learn under subpopulation shift. Specifically, it assumes multi level labels are available in the dataset, and train the neural network models with multiple prediction heads on each level of the labels. Empirical results on multiple datasets show that this training paradigm performs better than baselines especially when there is explicit subpopulation shift in the test data.



**Requested Changes:**

1. Since the results in this paper are aggregated from multiple random runs, please include the standard deviation in the results. In some cases, the difference in the absolute numbers between methods are quite small, and it is not very easy to tell how significant the improvements are.

2. The usefulness of the proposed methods is not very convincing since the improvements (esp. comparing to BCNN-18) are usually very small, in both accuracy and the proposed *catastrophic coefficient* metric. For example, in Section 4.3, "10.0% improvement" was observed in catastrophic coefficient, which is 2.7 vs 2.43. The gap is even smaller comparing to the stronger baseline (2.53). It is a bit hard to interpret the gaps, but considering the performance gap between the case of Seen->Seen and Seen->Unseen is 2.7 vs 0.89, it is clear that the proposed algorithm is not narrowing this gap by much.

3. As mentioned in 2, the proposed method utilizing multi-level labels does not seem to close the performance gap caused by subpopulation shift by much, even comparing to the simple baseline that just use flat class labels. In practice, collecting multi-level hierarchical labels could be more complicated than simple labels. I think this paper could be improved by providing more evidence on how the extra efforts spent on collecting multi-level labels are worth it. Moreover, it could discuss how to extend the proposed algorithm to the case where no multi-level labels are readily available.

4. Although this paper focuses on subpopulation shift, it should not ignore the literature on more general distribution shift. The related work section could be expanded to include this area, and more importantly, the empirical studies should be extended to include baseline algorithms that were designed to deal with general distribution shift.

5. Even if the proposed algorithm does not significantly improve over the baseline at closing the gap caused by subpopulation shift, it could still be interesting for this venue as an empirical study. However, this paper is currently missing some important analysis on this aspect.

    * Systematic studies and results on the impact of where to put different prediction heads are needed.
    * Ablation studies regarding different training strategies (i.e. conditional or non-conditional) are needed.



**Strengths And Weaknesses:**

## Strength

* A new training algorithm for learning with multi-level labels with conditional training.
* A new study of how training with multi-level labels could potentially help with learning under subpopulation shift.

## Weakness

The experiment results are weak and could not support that the proposed methods is very helpful for fighting against subpopulation shift. Please see "request changes" below for more details.

---

> ### Author Response · Authors · 2022-08-16
> **Rebuttal for Reviewer Z8z2**
>
> Thank you for the review. We have incorporated the suggestions, and elaborate on some here.
>
>
>
> 1. In the original draft of the paper, we had run our experiments for three runs and reported the mean numbers. We have now run it for 5 iterations, and we have updated the draft with mean and standard deviation of all 5 runs. The main text is updated with the new means, and the standard deviations are added to the appendix.
>
>
>
> 2. We have updated the catastrophic coefficients improvements in the text to be absolute changes rather than percentage changes, in order to make the improvements clearer. We wish to emphasize that in this paper, we are using the catastrophic coefficient only as an evaluation metric to see the effect of incorporating hierarchy instead of explicitly optimizing for it, and hence we do not see very big changes. We still hope that we are able to highlight the fact that with very minimal changes, and not exposing the evaluation metric or the shifted subpopulation to the training methodology, incorporating hierarchies still results in improvements, albeit slight in the problem resulting from subpopulation shift.
>
>
>
> 3. In many cases, the dataset is built using an underlying hierarchy such as WordNet for ImageNet. In these cases, the labels are available for free. In cases where the label hierarchies are not known, independent and orthogonal works such as [1] can be used to build the hierarchies at some additional cost.  We have added this to the relevant prior works section as well. In general, there are not many works on subpopulation shifts that show large improvements, except for papers that involve access to the shifted dataset for semi-supervised training such as [2] where the authors have applied the idea behind the original [3] paper. However, being semi-supervised, it assumes access to the unlabeled unseen shifted distributions, which is not often practically viable. Hence, we hope that the small improvement itself shows that there is information in hierarchies that can help the problem of subpopulation shift in cases without exposure to the shifted subpopulations themselves.
>
> [1] Fast and Balanced: Efficient Label Tree Learning for Large Scale Object Recognition.
>
> [2] A Theory for Label Propagation for Subpopulation Shift
>
> [3] FixMatch : Simplifying Semi-Supervised Learning with Consistency and Confidence
>
>
>
> 4. We have included more literature on general domain adaptation techniques (in the appendix, due to space constraints). Using general domain adaptation techniques for the specific problem of subpopulation shift is an interesting area by itself and we thank the reviewer for the pointer.
>
>
>
> 5. Thank you for the pointer to further improve our experiments and discussion section. We have not performed ablation studies for conditional vs non-conditional training, primarily because the difference between them is essentially captured by the difference in the results between the Branch-CNN paper and our method. Branch-CNN is a hierarchical setup similar to ours, with multiple heads but the only interaction between them occurs at the loss. In our case, the validity matrix allows conditional propagation of labels. We show improvements over Branch-CNN primarily because of this conditional setup. We hope that that serves as a form of an ablation study, and if the reviewer finds that it will be helpful to include this discussion in the draft, we’d be happy to update it.

---

### Review · Reviewer_iNPr · 2022-07-22

**Summary Of Contributions:**

This paper studies the important problem of distribution shift.  In particular, this paper studies subpopulation shift. This paper views the the problem of subpopulation shift through the lens of hierarchical learning.  In hierarchical learning, the labels follow a hierarchy. For example, the label Tiger Cat (sbuclass) is the child of Felidae (class), and Felidae is the child of Mammal (superclass).

Below are the contributions I see in this paper:

1. they proposed a new metric called “catastrophic co-efficient” which penalizes misclassification at a higher hierarchy level more than misclassification at the lower level.
2. they proposed a new network architecture and loss to incorporate the hierarchical structure of the labels.
3. experimentally, their method shows better accuracy on an existing benchmark, BREEDS [1].  (While they also have 2 custom datasets, I don’t see them as significant additions to BREEDS, and the authors didn’t explain clearly why these 2 custom datasets are significant additions.)

[1] Shibani Santurkar, Dimitris Tsipras, and Aleksander Madry. {BREEDS}: Benchmarks for subpopulation shift. In International Conference on Learning Representations, 2021.

**Broader Impact Concerns:**

The authors did not discuss broader ethical impacts, but I also don’t foresee any.

**Requested Changes:**

Please see the weaknesses in my response above.  As of the current draft, I don’t see any marginal modification that will change my mind.  The fundamental issue is that I am not convinced that hierarchy in a feedforward network corresponds to the hierarchy of the labels.  I do like the perspective of hierarchical learning, so perhaps another iteration of the method will make a strong submission.

**Strengths And Weaknesses:**

# Strength —
- This paper studies an important and new problem.
- The perspective of using hierarchical learning to handle subpopulation shift is nice.

# Weakness —
- Here I outline my major concerns:
    - “Earlier layer representation corresponds to higher level in the label hierarchy”.  This seems to be the assumption behind the proposed method, but I’m not convinced that this is true.  I think earlier layers in a network corresponds to lower level statistics of an image, but that is not necessarily enough for classifying at the superclass level.
    - “Availability of subpopulation label?”  Figure 2 suggests that there are losses at all 3 levels: superclass, class, and subclass.  In this case, a label at the subclass label needs to be provided.  In page 6, it says “The models only see labels till the class level; the **subpopulations labels are hidden from the networks.**” This contradicts with my understanding.  I think assuming access to subpopulation label is potentially an unfair advantage, and would like the authors to clear this confusion.
    - “What is the validation set?”  The hierarchical network introduces a few more hyperparameters.  For example, where the classification heads are inserted in a ResNet backbone is a design choice.  The author wrote this was done experimentally, but how was performance measured during validation?  Did the model have access to the target distribution?  If we did this again using a different backbone, e.g. larger ResNets, would we still get good results?  Assuming access to the target distribution for validation also seems to like a potentially unfair advantage.
- Overall, combining my major concerns, I cannot confidently say that the proposed method is a solid improvement in addressing the subpopulation shift problem.
- In addition, I think the writing can be significantly improved.  Here I show only a few examples:
    - In the introduction, when the class labels are discussed, many of them have hyphens, ‘-’, that are unnecessary and confusing.  Are they supposed to be in plain text, or some type of symbol in programming language.  Some labels are capitalized and some not.
    - Figure 2 and 3 overlap a lot, why not just combine them into a single figure?
    - In Figure 3, the notations used in the legends are not consistent with what’s used in the main text.
    - The main text is incomplete without the figures.  Section 3.5 relies on the caption of Figure 2.  Both should be independently comprehensible.
    - The mask used for training is sometimes called the “validity matrix”, and other times “validation matrix”.  Fix the inconsistency. Also, it’s a vector, not a matrix.  I think it should be named simply as label mask.
    - This sentence in Section 4.2 is incomprehensible: “The method overseas training via a weighted summation loss where each contribution comes from each branch (head) of a multi-headed network.”
- As a suggestion, why not normalized the catastrophic coefficient by the max depth of the hierarchy.  This will make comparing across datasets easier.

---

> ### Author Response · Authors · 2022-08-08
> **Rebuttal for Reviewer iNPr**
>
> Thank you for the detailed review. In this response, we address the three weaknesses mentioned and want to clarify these points. We thank the reviewer for the comments on the writing and will be sure to fix these in the updated draft before the end of the rebuttal period.
>
>
> 1. “Earlier layers corresponding to higher level in the label hierarchy”
>
> A:  The reviewer is indeed correct that the earlier layers learn general low-level features. However, we wish to point out that our earliest head (corresponding to the highest level in the hierarchy) is after 3 out of 4 ResNet blocks in the ResNet18 architecture. This means that we make classification at the 14th layer out of a total of 18 layers, which gives the network enough representational capacity to learn high level class-relevant  features. We apologize for not making this clearer in the text and will remedy that in our final draft before the end of the rebuttal period.
>
> The idea is similar to anytime inference utilized in neural networks as early exit conditions, as shown in [1]. The premise is that simpler examples do not need to traverse the entire depth of the architecture and can be inferred upon in the earlier layers. In our case, the “simplicity” of the classes in the higher levels of the hierarchy is coming from two facts: a) early levels correspond to more visually different categories than the deeper levels of the hierarchy, b) the number of classes to distinguish between increases with the level of the hierarchy, and since early layers have lesser classes to separate, it becomes a relatively simpler task than at the later layers.
>
>  Our hope is that this addresses the comment in requested changes that “fundamental issues is that [..]. not convinced that hierarchy in a feedforward network corresponds to the hierarchy of the labels”. Indeed, it is possible that the visual hierarchy learned by neural networks usually does not adhere to the semantic label hierarchy. Our hope with the conditional training framework is to enforce this hierarchy not just in the features learned, but also in the form of inference that enables better inference and lesser catastrophic mispredictions under subpopulation shift.
>
> [1] BranchyNet: Fast Inference via Early Exiting from Deep Neural Networks
>
>
>
> 2. “Availability of subpopulation label?”
>
> A:  We apologize for the confusing wording here. We never show subpopulation labels to the network. They are only used for evaluation. In Figure 2, we have shown three levels of our population and for easier internalization, we called each level “superclass”, “class” and “subclass”. In this case, the subpopulation is one level below “subclass” and is not shown in the figure, since the network does not get access to labels at that level. The final classification class in that figure will be “Dog” vs “Cat” for both seen and target cases, but the breeds of “Dog” and “Cat” that make up the subpopulations will shift. Figure one is taken from the custom dataset and Figure 2 outlines an example taken from the hierarchy in BREEDS’ LIVING-17 dataset, and the depth of each of these are outlined in Table 1. We will make this clearer in our final draft.
>
>
> 3.  “What is the validation set?”
>
> A: Thank you for the super relevant question. Indeed, we have four sets available to us. The training and validation sets of seen distributions, and the training and validation sets of the unseen distributions. For training, we use the training set of the seen distribution, and for validation, we use the validation set of the seen distribution itself. While validating on the unseen distribution might have given us better results, the aim was to try and figure out if hierarchical distributions help in subpopulation shift, and hence the shifted distributions are only used as test set, rather than for validation. The final results are reported on the validation set of the unseen distribution, as reported by BREEDS. This is the same reason that we did not optimize the catastrophic coefficient directly as part of the loss function and rather used it only as an evaluation metric. This is a great point brought out, and we will be sure to make this clearer in the updated draft.

---

### Review · Reviewer_emQ8 · 2022-08-02

**Summary Of Contributions:**

Subpopulation shift is an issue for generalization in which the distribution of classes changes during test time and unseen classes may appear.
This work highlights the independence of classes in standard training as a potential cause, that is, it highlights how common losses assign equal weight to all misclassifications.
For many problems of interest, the classes may have some dependence, and in particular relate to each other hierarchically by sharing coarser classes.
This work proposes a structured labeling scheme to incorporate hierarchical labeling during training for improved robustness to subpopulation shift.
In this scheme each image is assigned a hierarchy of labels, and a corresponding hierarchy of predictors or "heads" is attached to a deep network backbone (such as a standard residual network) where each label level is handled by its corresponding predictor level.
These predictors are cascaded in the sense that an incorrect classification at one level skips training of the sublevels.
As a measure of improvement, a measure of "graphical distance" is introduced to score misclassifications according to hierarchical labeling, and a normalized variant called the "catastrophic coefficient" is also measured.
These are similar to lowest common ancestor (LCA) metrics used by prior work on hierarchical classification, which measure the level at which a true and predicted class pair meet in the label hierarchy.
Experiments on subpopulation shift benchmarks, established by the prior work BREEDS, show improvement in standard accuracy and the proposed graphical distance on seen (training subpopulations) and unseen (testing subpopulation) data.
For accuracy the improvements tend to be in the range of 1-2 absolute percentage points.

**Broader Impact Concerns:**

This work does not have a broader impacts section and it does not need one.

As the topic is robustness to shift, in particular sensitivity to different subpopulations of the data, the potential ethical impact of the work is in fact positive.
Models that cope poorly with subpopulation shift are liable to make unfair predictions that disproportionately affect certain classes, especially those classes which may not have been well represented in the training set (if they were present at all).
Reducing error in the presence of this type of shift should only help with the deployment of a model for any ethical use case of machine learning.

**Requested Changes:**

- Please discuss the connection between source acuracy (without shift, that is the "s-s" condition in this work) and subpopulation shift accuracy.
  It seems that these are mostly correlated, except for a few cases of ~1 point discrepancies, so is it necessary to specifically train for subpopulation shift, or will generally better classifiers resolve the issue?
  I expect specialized work for shift could do better, as pursued in this work, but the results and their importance would be clarified by noting this trend and discussing it in the results section.
- The related work could have better coverage of hierarchical classification and adaptation (please see references in this section and the previous section of the review).
  In particular, it is necessary to give further credit to prior study of super-to-sub category inference, as done by Label Trees for example.
- The citation of Goodfellow 2016 for domain adaptation is overly generic and ahistorical. Consider citing more primary and older references, such as [A, B, C], as research references to go complement the textbook citation.
- The introduction on domain adaptation and shift could better situate the work w.r.t. related types of shift and existing terminology. For instance, "label shift" [D] or "target shift" [E] have been used to indicate shifts not of the input, as is the focus of most of domain adaptation, but instead shifts in the marginal distribution of labels, or which classes are even present/absent.
- "Graphical" distance may be misinterpreted as a name. Please consider "tree" or "hierarchy" distance instead.

[A] Dataset shift in machine learning. Joaquin Quionero-Candela, Masashi Sugiyama, Anton Schwaighofer, and Neil D Lawrence. MIT Press, Cambridge, MA, USA, 2009

[B] Adapting visual category models to new domains. Kate Saenko, Brian Kulis, Mario Fritz, and Trevor Darrell.  In European conference on computer vision, pp. 213–226. Springer, 2010

[C] Unsupervised domain adaptation by backpropagation. Yaroslav Ganin and Victor Lempitsky. In ICML, 2015

[D] Domain Adaptation with Conditional Distribution Matching and Generalized Label Shift. des Combes et al. NeurIPS'20.

[E] On Target Shift in Adversarial Domain Adaptation. Li et al. AISTATS'19.



**Strengths And Weaknesses:**

Strengths

- The chosen topic is a real issue. Subpopulation shift happens and it hurts model accuracy, and does so in a potentially biased fashion due to the presence/absence of some subclasses within a superclass.
  The hierarchical labeling of super/sub-populations is well illustrated in Figure 1.
- The method is explained clearly and practical implementation choices are made, such as the masking the losses (Figure 3) for simplicity.
- The choice of benchmarks is sound.
  BREEDS [Santurkar et al. 2021] is derived from large-scale image data, like ImageNet, by selecting subclasses from superclasses and dividing them between the training and testing sets.
  This prior work took care to identify visually coherent superclasses and adjusted the ImageNet hierarchy inherited from WordNet accordingly.
- The claimed contribution of evaluating current networks and subpopulation shift benchmarks (Section 1, third bullet point, pg. 3) is substantiated by considering the standard BREEDS benchmark and a small custom dataset with standard ResNet architectures for image classification (ResNet-18, for a small network, and ResNet-50, for a larger network).
- This work reinforces a related point made by [Bertinetto et al. 2020] that not all hierarchies and not all subpopulations are alike (see Section 4.3 of this work). There are easier and harder shifts among subpopulations. This argues for reporting results across multiple shifts, and not simply aggregating across them and reporting average accuracy.
  The experiments confirm this with multiple dataset variants with different choices of subpopulations.
  Regardless of other experimental issues (see Weaknesses below), these are informative results on the granularity and particularity of subpopulation shifts.
- Experiments are repeated across multiple seeds to check for sensitivity to orderings of the data and parameter initialization (and results are robust to these factors).

Weaknesses

- The claimed contribution of conditional training by filtering (Section 1, first bullet point, pg. 3) is weakened by many works on cascading models and filtering inputs like [I]. As a related but more distant example, anytime inference for image classification has also studied coarse-to-fine/super-to-sub class recognition as in [Anytime Recognition of Objects and Scenes. Karayev et al. CVPR'14].
- The claimed contribution of introducing misprediction impact (Section 1, second bullet point, pg. 3) is weakened by work on hierarchical error [II, Bertinetto et al. 2020] for deep net training and the original ImageNet papers like [IV].
  This contribution is further tempered by the existence of "cost-sensitive" classification as a more general topic, of which hierarchical cost is a special case.
- The experiments are lacking in comparisons to hierarchical supervision baselines from existing work at ImageNet-scale, although such works are cited [Bertinetto et al. 2020, Deng et al. 2014].
  Branch-CNN [Zhu & Bain, 2017] is the only such baseline considered, and it is neither as recent [Bertinetto et al. 2020] or as high-profile as an award-winning paper [Dent et al. 2014].
  To the best of my understanding, the Bertinetto method should likewise be computationally efficient, and applicable to these datasets.
  Only comparing to standard ResNets with and without the proposed hierarchical training is not as informative as evaluating against existing work, if the research question is indeed whether or not hierarchical training helps with subpopulation shift.

Miscellaneous Feedback

- Please do a general editing pass for spelling and grammar. For example, the proposed catastrophic coefficient is misspelled as "catastophic" on pg. 2. In the list of contributions on pg. 3, there are too many commas, as "To the best of our knowledge," would be fine.

[I] Fast and Balanced: Efficient Label Tree Learning for Large Scale Object Recognition. Deng et al. NeurIPS'11.

[II] Learning hierarchical similarity metrics. Verma et al. CVPR'12.

[III] Making Better Mistakes: Leveraging Class Hierarchies with Deep Networks. CVPR'20.

[IV] Deng et al. Imagenet: A large-scale hierarchical image database. CVPR'09.

---

> ### Author Response · Authors · 2022-08-12
> **Rebuttal for Reviewer emQ8**
>
> Thank you so much for eloquently expressing a concise summary of what we really hoped to convey with our draft in the “strengths” section. Would you mind if we used some of the points mentioned in the strength along with the verbiage used by the reviewer to update our draft, with acknowledgements to the anonymized reviewer?
>
> We are fixing the suggested citations and changes in the section on requested changes, and will have it rectified by the final draft. Here, we attempt to address some of the weaknesses.
>
>
>
> 1. Regarding anytime inference and misprediction impact contributions:
>
> A: Thank you for pointing out the relevant body of research on anytime inference and cost-sensitive misclassification we have missed on citing. This will be rectified in the final draft. We agree that these are indeed well-established areas of research, however we believe that its impact on the problem of subpopulation shift has not yet been studied. The aim with anytime inference traditionally is to result in more efficient inference pipelines. In contrast, in our method we use the conditional training via filtering methodology to enable the networks to propagate conditional probabilities that aid in making better decisions under subpopulation shift. We concede that the formulation is not novel, rather the usage is, and will clarify this in the contributions section of the updated draft. With respect to the misprediction impact, our goal was to give a holistic assessment to show that with hierarchies, the network tends to get lesser examples wrong, and with the examples it does get wrong, they are 'closer' to the ground truth, and we used graphical traversal as the measure of 'closeness'. We will clarify this in the contributions and milden the claim to mention that we employ known strategies to show this behavior.  We also wish to emphasize that unlike Bertinetto et al. 2020, we only use this as an evaluation metric to see the impact of hierarchies on subpopulation shift, and do not optimize it during training.
>
>
>
> 2. Regarding S-T accuracy tracking S-S accuracy:
>
> A: Thank you for the relevant question. We noticed that this was a trend that was not unanimous with different label hierarchies and therefore did not make generalized comments on it. For instance, improvements in S-S and S-T accuracies track each other roughly in both custom datasets and LIVING-17 datasets, but not for non-LIVING-26 and ENTITY-30 datasets. We believe that the kind of shifted subpopulation itself has an impact on this, as evidenced by the different results we get by shifting one source to three targets in LIVING17-A, B and C. We also believe that the classes where the shift occurs determine how easy the categorization under shift is, and hence see different trends for say, LIVING-17 and Non-LIVING-26 datasets. However, as the reviewer pointed out, this discussion does add value and we will make a note of this in the revised draft, without making generalized comments.

---

> > ### Comment · Reviewer_emQ8 · 2022-09-01
> > **Thank you for the response.**
> >
> > Thank you for the response. The revision is an improvement, but I am still concerned about the breadth of experiments and depth of the contributions relative to existing work, specifically in terms of sufficiently measuring the effect of hierarchical learning on subpopulation shift.
> >
> > > Would you mind if we used some of the points mentioned in the strength along with the verbiage used by the reviewer to update our draft, with acknowledgements to the anonymized reviewer?
> >
> > Please use the text of the review as you see fit while crediting the TMLR review process. At least, I do not see an issue with using the content of the open reviewing process provided proper scholarly attribution.
> >
> > > Regarding anytime inference and misprediction impact contributions
> >
> > Thank you for clarifying the focus and novelty of this work. WIth the correct credit to existing formulations and methods, this work can still contribute informative experiments, which are themselves made more clear by identifying the methods employed and where they have come from so that an interested reader can learn more.
> >
> > The revision better highlights prior work, but it does not engage with existing results by comparison. By not experimenting more fully with hierarchical learning, the contributions regarding (1) training with hierarchical learning and (2) quantifying the misprediction impact are not completely delivered. Reviewer Z8z2 and I have both commented on the need for systematic study and comparison with other methods, and I still find the lack of experiment with Bertinetto et al. 2020 concerning. It is evident that this work only uses the given hierarhical scoring methods for evaluation, and not training, but comparing the proposed conditional training with their unconditional training would provide a more thorough and informative study of how "to tackle the problem of subpopulation shift by hierarchical learning methods" which is the aim of the first and main contribution.
> >
> > > Regarding S-T accuracy tracking S-S accuracy
> >
> > Thank you for further unpacking these results and scoping these comments to the specific shifts investigated without overgeneralizing to other data.

---

### Decision · Action_Editors · 2022-09-17

**Recommendation:** Reject

**Comment:**

This submission aims to improve robustness to subpopulation shift using a hierarchical training method that adds auxiliary heads to intermediate network layers to classify different levels of the hierarchy. The improvement in accuracy the method provides is marginal, but it appears to be statistically significant for most of datasets investigated when evaluating under subpopulation shift. However, the reviewers pointed out that the paper does not compare with previous baselines for leveraging the label hierarchy during training or improving robustness to distribution shift. Although I agree with the authors that the goals of previous work leveraging the label hierarchy are different from the goals of this work, the methodology is similar and thus the baselines are relevant. Reviewers also raised questions regarding the rationale behind classifying higher levels of the label hierarchy from earlier layers of the network, and asked for further study of the placement of the classifier heads and ablations regarding the choice of loss function, which were not provided during the rebuttal period.

Although evaluation of hierarchical learning methods for subpopulation shift is a topic that could be of widespread interest, the empirical justification and evaluation of the proposed method in the current submission is insufficiently convincing. I hope the authors will consider resubmitting their paper to TMLR after performing the experiments requested by the reviewers.